# MMP14 Contributes to HDAC Inhibition-Induced Radiosensitization of Glioblastoma

**DOI:** 10.3390/ijms221910403

**Published:** 2021-09-27

**Authors:** Yuchuan Zhou, Hongxia Liu, Wang Zheng, Qianping Chen, Songling Hu, Yan Pan, Yang Bai, Jianghong Zhang, Chunlin Shao

**Affiliations:** Institute of Radiation Medicine, Shanghai Medical College, Fudan University, Shanghai 200032, China; 18211140009@fudan.edu.cn (Y.Z.); 18111140003@fudan.edu.cn (H.L.); 19111140004@fudan.edu.cn (W.Z.); 17111140001@fudan.edu.cn (Q.C.); 18111140002@fudan.edu.cn (S.H.); swallowpan@fudan.edu.cn (Y.P.); yangbai@fudan.edu.cn (Y.B.)

**Keywords:** GBM, MMP14, radioresistance, HDAC inhibitor, SAHA, bioinformatic analysis, RNA-seq, ChIP-seq

## Abstract

Glioblastoma (GBM) is the most common and malignant primary brain tumor in adults. Radiotherapy has long been an important treatment method of GBM. However, the intrinsic radioresistance of GBM cells is a key reason of poor therapeutic efficiency. Recently, many studies have shown that using the histone deacetylase (HDAC) inhibitor suberoylanilide hydroxamic acid (SAHA) in radiotherapy may improve the prognosis of GBM patients, but the underlying molecular mechanisms remain unclear. In this study, Gene Expression Omnibus (GEO) datasets GSE153982 and GSE131956 were analyzed to evaluate radiation-induced changes of gene expression in GBM without or with SAHA treatment, respectively. Additionally, the survival-associated genes of GBM patients were screened using the Chinese Glioma Genome Atlas (CGGA) database. Taking the intersection of these three datasets, 11 survival-associated genes were discovered to be activated by irradiation and regulated by SAHA. The expressions of these genes were further verified in human GBM cell lines U251, T98G, and U251 homologous radioresistant cells (U251R) by reverse transcription-quantitative polymerase chain reaction (RT-qPCR). It was found that MMP14 mRNA was considerably highly expressed in the radioresistant cell lines and was reduced by SAHA treatment. Transfection of MMP14 siRNA (siMMP14) suppressed cell survivals of these GBM cells after irradiation. Taken together, our results reveal for the first time that the MMP14 gene contributed to SAHA-induced radiosensitization of GBM.

## 1. Introduction

Glioblastoma (GBM) is the most common and aggressive primary brain tumor in adults [1]. Despite considerable advances in fundamental research on GBM that have been achieved in recent years, the median survival time for GBM patients following conventional therapy is only approximately 15 months [2]. For a long time, radiotherapy has been a major treatment strategy for GBM. However, the intrinsic radioresistance of GBM cells is the main reason for unsatisfactory therapy, and the expression of inherent radioresistance genes may be adversely correlated with the survival of GBM patients. Due to the poor prognosis of GBM patients and the limited effectiveness of traditional radiotherapy, there has been a steady increase in novel radioresistance-related biomarkers and radiosensitizers over the last decade.

Histone deacetylase inhibitor (HDACi), a tiny compound that can increase histone acetylation levels by inhibiting the activity of histone deacetylase (HDAC) [3], has the potential to increase tumor radiosensitivity [4,5]. HDACs are enzyme families whose enzymatic activity controls the acetylation state of protein lysine residues [6]. Acetylation of histones influences gene expression via influencing chromatin structure [4,7]. Inhibition of HDACs can activate antitumor pathways by inducing cell growth arrest, terminal differentiation, cell death, and autophagy [8,9]. The ability of HDACi to induce DNA damage may be increased when used in conjunction with radiation [2,10,11]. For example, the addition of valproic acid, a type of HDACi, to radiotherapy in combination with temozolomide improved the outcomes of primarily diagnosed GBM patients [10]. HDACi has been proven to be a versatile antineoplastic drug because of its involvement in the control of cell cycle, apoptosis, DNA damage response, metastasis, autophagy, and other cellular processes [4,11,12]. Suberoylanilide hydroxamic acid (SAHA), commonly known as vorinostat, was the first clinical HDACi drug authorized by the US Food and Drug Administration [13], and it offers an essential therapeutic method for GBM [14]. In some patients, the application of SAHA improved the efficiency of regular chemoradiotherapy [11].

However, the molecular targets contributing to SAHA-induced radiosensitization of GBM remain unclear, which may become a barrier to the therapeutic application of SAHA. This study employed the Gene Expression Omnibus (GEO) database, the Chinese Glioma Genome Atlas (CGGA), and the Gene Expression Profiling Interactive Analysis (GEPIA) database to comprehensively investigate the molecular biomarkers of GBM radioresistance to identify novel targets of SAHA in GBM therapy.

Bioinformatics analysis in this study can be divided into three sections. First, the GEO dataset GSE153982 was examined to assess radiation-induced changes in gene expression in GBM. Second, we searched the CGGA database for genes associated with survival. Third, the GEO dataset GSE131956 was analyzed to screen genes whose expression may be altered after SAHA treatment. Following these methods, 11 survival-associated genes that were both regulated by SAHA and activated by radiation were identified. Next, we analyzed the RNA expression levels of these candidate genes in three human GBM cell lines and discovered that the matrix metallopeptidase 14 (MMP14) gene was the most compatible gene with the results of bioinformatics analysis. MMP14 governs the formation and principal functions of invadopodia and plays an important role in cell migration and invasion [12]. According to previous research, MMP14 was regarded as a potential diagnostic or prognostic marker in multiple types of cancer [15,16,17,18]. Therefore, targeting MMP14 represents a novel approach to inhibit the migration and invasion of cancer cells [19,20]. However, there are currently few drugs developed against MMP14, and the roles of MMP14 in the radiosensitivity of GBM have rarely been reported. In this study, we discovered that SAHA could sensitize GBM cells to radiation by inhibiting MMP14 expression, which provides an experimental basis for the application of SAHA in GBM treatment.

## 2. Results

### 2.1. Evaluation of Radiation-Induced Changes in Gene Expression

To assess the changes in radiation-induced gene expression in GBM, we analyzed the GEO dataset GSE152982, which contains RNA-seq data generated from patient-derived GBM cells treated with 4 Gy radiation. RNA samples with three replicates from this dataset were extracted 48 h after irradiation. A cluster analysis of differentially expressed genes (DEGs) between irradiated and non-irradiated cells was performed (Figure 1A), and a volcano plot was displayed (Figure 1B), which showed that the gene expression patterns of these two groups were significantly different. In total, 428 genes were upregulated, and 588 genes were downregulated in irradiated cells compared with non-irradiated cells (Figure 1C). A Kyoto Encyclopedia of Genes and Genomes (KEGG) pathway enrichment analysis of DEGs was conducted, and the top ten KEGG enrichment pathways are shown in Figure 1D. Notably, many DEGs were involved in the cell cycle and p53 signaling pathways. In addition, we performed gene set enrichment analysis (GSEA) to functionally annotate the gene expression differences between irradiated and non-irradiated cells and revealed that the transcriptional signature of DEGs was enriched in the pathways of microtubule cytoskeleton organization, mitotic cell cycle, mitotic sister chromatid segregation, nuclear division, and sister chromatid segregation (Figure 1E). Because cell cycle arrest and DNA damage repair are the main biological responses after irradiation [21], the results of RNA-seq analysis appear to be reliable.

### 2.2. Screening of Survival-Associated Genes

GBM is classified as a grade IV glioma according to World Health Organization (WHO) definitions [22]. To identify survival-associated genes in GBM, we analyzed 249 high-grade glioma samples with WHO grade IV in dataset “mRNAseq_693” from the CGGA. Subjects with an overall survival <30% (overall survival <8.5 months) were classified as having short-term survival, whereas those with an overall survival >70% (overall survival >20 months) were classified as having long-term survival. DEG analysis of the two groups was performed using the Bioconductor package limma. Totally, 143 DEGs were identified, including 140 upregulated genes and 3 downregulated genes (Figure 2A). According to the KEGG pathway enrichment analysis, the survival-associated DEGs were strongly connected to the pathways of focal adhesion, phosphatidylinositol-3 kinase (PI3K)-Akt signaling pathway, extracellular matrix (ECM)-receptor interaction, and regulation of actin cytoskeleton pathways (Figure 2B). A chord plot was created to provide an overview of the relationships between the genes linked to multiple processes (Figure 2C). DEGs annotated to specific Gene Ontology (GO) terms could be identified using this representation. For example, MMP14, a significantly upregulated gene, was involved in the pathways of extracellular matrix organization, extracellular structure organization, and cell–substrate adhesion processes. A total of 143 DEGs were assigned to 50 sub-categories of GO terms from three main categories: cellular component (CC), biological process (BP), and molecular function (MF) (Figure 2D). The top two sub-groups that contained most genes in the BP group were extracellular matrix organization (GO:0030198) and extracellular structure organization (GO:0043062). The top sub-group that contained most genes in the CC group was the collagen-containing extracellular matrix (GO:0062023). Moreover, the terms of extracellular matrix structural constituent (GO:0005201) and growth factor binding (GO:0019838) were associated with the greatest numbers of genes. The z-score was automatically computed using the Bioconductor package GOplot, which indicated whether a GO term was reduced (negative value) or increased (positive value). These analyses effectively limited the scope of the subsequent gene screening.

### 2.3. Analysis of Histone Acetylation-Controlled Genes

Histone acetylation plays an essential role in gene transcriptional regulation, and histone H3 lysine 27 acetylation (H3K27ac) is an epigenetic mark for active regulatory regions [23,24]. Therefore, we hypothesized that the expression of genes with H3K27ac gain at their promoters may be dramatically changed after SAHA treatment. To identify the potential genes controlled by histone acetylation, we downloaded the raw data of GEO dataset GSE131956, which contains H3K27ac chromatin immunoprecipitation-sequencing (ChIP-seq) of GBM [25]. The input control signal was subtracted from the ChIP signal before plotting. Annotating the characteristic peaks with ChIPSeeker [26], we found that 28.80% of the peaks were in the promoter regions (Figure 3A).

In addition, an enrichment of H3K27ac ChIP peaks around the transcription start sites (TSSs) was observed (Figure 3B,D). After the annotation of peaks, 5474 genes with H3K27ac peaks at their promoters were extracted as candidate genes, and the top ten pathways of KEGG enrichment analysis are shown in Figure 3C. These candidate genes were closely related to Alzheimer’s disease, amyotrophic lateral sclerosis, and Huntington’s disease. Significantly enriched GO terms were also analyzed. Figure 3E illustrates that the transcription coregulator activity (GO:0003712) had the greatest number of genes in the MF group ribonucleoprotein complex biogenesis (GO:0022613), that the ncRNA metabolic process (GO:0034660) had the highest number of genes in the BP group, and that the cell–substrate junction (GO:0030055) had the greatest numbers of genes in the CC group. Genes whose expression may be altered as a result of SAHA treatment were identified based on these analyses.

### 2.4. MMP14 Was the Potential Target of SAHA

To learn more about the major potential biomarker genes contributing to SAHA-induced radiosensitization of GBM, the intersection of the above three datasets was analyzed and 11 genes were obtained (Figure 4A). “Radio_genes” indicates the DEGs after irradiation, “CGGA_genes” represents the survival-associated genes, and “ChIP_genes” represents the candidate genes that may change after SAHA treatment. H3K27ac was found to be highly enriched in the promoters of these 11 genes (Figure 5). We hypothesized that these 11 genes may contribute to SAHA-induced radiosensitization in GBM.

To verify the bioinformatic analysis results, the in vitro experiments were performed using human GBM cell lines (U251 and T98G) and U251 homologous radioresistant cells (U251R). U251 and T98G cell lines with various radiosensitivities are widely used in the study of human GBM [27]. The U251R cell line was previously established from U251 using fractionated irradiation of X-rays in our laboratory, which showed an increased survival after irradiation in comparison with U251 [28]. These data suggested that the highly radioresistant cells might represent a poor clinical prognosis. It was found that T98G cells had the highest radioresistance among these cell lines, followed by U251R and U251 cells (Figure 4B). To choose the best drug concentration of SAHA for further research, we evaluated the survival of these cell lines using the Cell Counting Kit-8 (CCK-8) assay at various concentrations and times (Figure 4C) and chose the drug treatment conditions of 5 μM for 24 h. Western blot analysis revealed that after this treatment of SAHA, H3K27ac protein expression was lower in untreated cells than in SAHA-treated cells (Figure 4D).

To further assess the potential involvement of the key candidate genes in the radiosensitivity of GBM cells, we examined the RNA expression levels of the key candidate genes in different GBM cell lines. Analysis of the CGGA datasets showed that RNA expression levels of 11 candidate genes were higher in the low-survival group (Table 1). Therefore, we hypothesized that the RNA expression levels of candidate genes would be higher in radioresistant cell lines. Consequently, the RNA expression levels of F3, IGFBP3, SOCS3, MMP14, and SPRY1 increased in U251, U251R, and T98G cells in an ordered manner (Figure 6A), which was positively correlated with the radioresistance of the three cells (Figure 4B). Furthermore, only GAL3ST4, SERPINE1, and PAMR1 expression was increased in T98G cells. However, there were no discernible changes in the RNA expression levels of PLAT, PBX3, PLAU, and SPRY1 in the three GBM cells.

After 24 h of 5 μM SAHA treatment, there was a substantial increase in the RNA expression levels of F3, GAL3ST4, IGFBP3, PLAT, and SERPINE1, and there was a significant decrease in the RNA expression levels of MMP14, PBX3, and PLAU in the three cell lines (Figure 6A). Moreover, there were no significant variations in the RNA expression levels of PAMR1 and SPRY1 in GBM cells after SAHA treatment (Figure 6B).

Furthermore, we verified the changes in the RNA expression levels of these candidate genes after irradiation (Figure 6C). Because the RNA samples in GEO dataset GSE153982 were extracted 48 h after 4 Gy irradiation [29], we treated GBM cell lines with the same dose in this study. After irradiation for 48 h, the RNA expression levels of F3 and MMP14 were increased in the three cell lines, which was consistent with the results of our bioinformatics analysis of “Radio_genes” (Table 1). The RNA expression levels of IGFBP3, SERPINE1, and PAMR1 were enhanced in the U251 and U251R cell lines. In U251R cells, the RNA expression level of GAL3ST4 was increased. PLAT levels decreased in U251R cells, whereas SOCS3 levels decreased in T98G cells. Furthermore, no significant changes in RNA expression of PBX3, PLAU, PAMR1, and SPRY1 were detected in the three cell lines.

According to the reverse transcription-quantitative polymerase chain reaction (RT-qPCR) data, MMP14 RNA expression was considerably higher in radioresistant cell lines and was reduced following SAHA treatment. In addition, the change in RNA expression level of MMP14 was compatible with the bioinformatics analysis of “Radio_genes”. As a result, we hypothesized that the decrease in MMP14 expression was involved in the radiation sensitizing effect of SAHA in GBM.

### 2.5. The Role of MMP14 in Radiosensitivity

MMP14 is highly expressed in numerous types of cancer, where it promotes angiogenesis, inflammation, cancer cell invasion, and metastasis [30]. Kaplan–Meier disease-free survival analysis showed that patients with high MMP14 expression in GBM had a significantly poorer prognosis than those with low MMP14 expression (Figure 7A) (data from GEPIA database). Therefore, we assumed that MMP14 was linked to radiosensitivity in GBM.

SAHA had an obvious radiosensitizing effect on GBM cells (Figure 7B–D). After SAHA treatment, the RNA expression level of MMP14 was significantly reduced (Figure 6B). Furthermore, the protein expression level of MMP14 was reduced in three GBM cells after SAHA treatment, as determined by Western blot analysis (Figure 7E). These results indicate that the application of SAHA and downregulation of MMP14 were largely related.

To further investigate the function of MMP14 in GBM, a small inferring RNA (siRNA) targeting MMP14 (siMMP14) and a scrambled small inferring negative control (siNC) were transfected into GBM cells. The transfection of siMMP14 significantly reduced the MMP14 levels in GBM cells compared with siNC (Figure 7F). When MMP14 was knocked down by MMP14 siRNA, the survival of U251, U251R, and T98G cells was significantly decreased after irradiation (Figure 7G). These results demonstrate that the decrease in the RNA expression level of MMP14 might be a reason for the radiosensitization of GBM cells with SAHA.

## 3. Discussion

GBM is a very aggressive tumor. Because of the limitations of standard therapy, a synergistic combination of therapies is required to counteract this aggressive brain tumor [31]. The drug development of radiosensitizer is an important link in the synergistic approaches to overcome current limitations of GBM treatment. HDACi, as represented by SAHA, is the most used radiosensitizing medicine, and it has also been used to treat GBM in conjunction with chemotherapy and radiation therapy [14]. HDACi can inhibit DNA repair responses and induce apoptosis, which contributes to enhanced sensitivity of tumor cells to chemotherapy and radiotherapy [32]. SAHA, for example, has been demonstrated to limit GBM development by reducing PRC2 function by decreasing EZH2 expression [33]. Furthermore, the application of SAHA suppressed c-Myc protein levels and extended animal survival in a patient-derived xenograft model [34]. However, the pharmacological targets of SAHA in the radiation sensitizing effect are poorly understood in GBM.

To identify the potential targets of SAHA for radiosensitization, we performed a series of bioinformatics analyses of GBM. Three databases were used to screen for potential targets. Containing RNA-seq data of the irradiated GBM cell line with three replicates, GEO dataset GSE153982 was analyzed to identify radiation-induced changes in gene expression. We believe that these “Radio_genes” are involved in the modulation of radiosensitivity in GBM. The CGGA database was used to screen for survival-associated genes to reduce the scope of the investigation. The intersection of the two groups was believed to be a possible radioresistant biomarker of GBM.

To further investigate the alteration of these candidate genes after SAHA treatment, H3K27ac ChIP-seq data (GEO dataset GSE131956) were analyzed. Depending on the gene target, histone acetylation marks can be present in the promoter regions where they can function by recruiting non-histone proteins to further influence gene expression [35]. Therefore, we hypothesized that genes with abundant H3K27ac marks in promoter regions would show notable alterations after SAHA treatment. Finally, the intersection of the three datasets yielded 11 potential genes. These 11 genes met three conditions, that is, their expression level changed after irradiation, their high expression was related to low survival of GBM, and their promoter regions had H3K27ac enrichment. Several previous studies have shown that IGFBP3, SOCS3, and MMP14 are related to radiosensitivity in certain types of cancers [36,37,38], which is consistent with our results that these three genes were highly expressed in radioresistant cells. Therefore, we assumed that these 11 genes were involved in SAHA-induced radiosensitization in GBM.

To verify the above conclusion, we examined the mRNA expression levels of 11 genes in three GBM cell lines. The results of RT-qPCR data confirmed that the RNA expression levels of these genes, except for PAMR1 and SPRY1, changed after SAHA treatment. Apart from PLAT, PBX3, and PLAU, the RNA expression levels of candidate genes were high in radioresistant GBM cell lines. Furthermore, the alterations in the RNA expression levels of F3, IGFBP3, MMP14, PAMR1, and SERPINE1 after irradiation were consistent with our bioinformatic study.

Since MMP14 is highly expressed in radioresistant cells and is decreased after SAHA treatment and since its expression trend after irradiation is compatible with the results of our bioinformatics study of GSE153982, we speculate that MMP14 is a possible target of SAHA. Based on GEPIA database analysis, a high MMP14 expression was associated with a poor prognosis in GBM patients. Some recent studies showed that the inhibition of MMP14 improves the efficiency of chemotherapy and radiotherapy by reducing the migration and invasion of cancer [20,36], which is consistent with the results in our study. Although MMP14 is an attractive target for cancer therapy, the development of drugs targeting MMP14 in vivo remains a difficult task.

Our study shows that the reduction in MMP14 can increase radiosensitivity in GBM, while the radiosensitizer SAHA can directly reduce the expression of MMP14. This indicates that the inhibition of MMP14 contributes to SAHA-induced radiosensitization in GBM. There exist several lines of evidence suggesting that SAHA and other HDACi have good application prospects as a radiosensitizer for GBM treatment [10,11]. The current study finds a novel function of SAHA in inhibiting the expression of MMP14 and provides basis for the clinical application of HDACi in the treatment of radioresistant GBM. However, the specific mechanism through which SAHA influences MMP14 expression requires further study. Although histone acetylation is typically associated with gene expression enhancement, there are instances where histone acetylation contributes to the suppression of gene expression. It has been shown that SAHA can block at or upstream of Akt in disrupting the PI3K/Akt pathway [39]. Additionally, the PI3K/Akt signaling pathway has been shown to activate MMP14 expression and enhance cell migration and invasion [40]. Therefore, we speculate that SAHA may repress the expression of MMP14 by blocking the PI3K/Akt pathway. Interestingly, histone acetylation, as it turns out, can also directly inhibit gene expression [41,42]. Some studies suggest that SAHA may restore the expression of a DNA damage repair gene called Ogg1 gene by modifying histone deacetylation and remodeling DNA methylation [43]. In other words, some epigenetic modifications caused by SAHA may inhibit MMP14 expression. Collectively, this study demonstrated that MMP14 was a key element in explaining the mechanism of SAHA treatment and provided an important direction for subsequent research.

## 4. Materials and Methods

### 4.1. Cell Culture and Irradiation

Human GBM cell lines U251 and T98G were purchased from the Cell Bank of the Chinese Academy of Sciences. The U251R cell line was previously developed from its parental cell line U251 by exposure to 2 Gy radiation/day for 30 fractions (5 fractions/week in general) with a total dose of 60 Gy [28]. U251, U251R, and T98G cells were cultured in DMEM (Gibco, Thermo Fisher Scientific, Waltham, MA, USA). All media were supplemented with 10% fetal bovine serum, 100 units/mL penicillin, and 100 mg/mL streptomycin (Gibco). Cells were incubated at 37 °C in 5% CO2 and subcultured every 3 days.

Cells in the log phase were irradiated with a dose rate of 0.883 Gy/min X-ray (X-RAD 320, PXI Inc., North Branford, CT, USA; 12 mA, 2 mm aluminum filtration) at room temperature.

### 4.2. SAHA Treatment

SAHA (SML0061, Sigma-Aldrich Inc., Shanghai, China) was freshly stocked in dimethyl sulfoxide (DMSO) at 5 mM. It was diluted into the medium of U251, U251R, and T98G cultures in the proper concentration, and the cultures were incubated without change of culture medium for proper hours. As the control of SAHA, U251, U251R, and T98G cells were incubated with 0.1% DMSO but without SAHA.

### 4.3. Colony Formation Assay

The radiosensitivity of U251, U251R, and T98G cells was assessed using a cell colony formation assay. Briefly, cells were plated in the six-well plates at a density of 150, 250, 450, 1000, and 2000 cells/well. After full attachment, they were exposed to 0, 2, 4, 6, and 8 Gy radiation. Approximately 2 weeks after radiation, cell colonies were fixed with methanol for 20 min and stained with 0.1% crystal violet for 30 min to count them. The number of colonies with more than 50 cells was determined. The colony formation assay was repeated three times for each cell sample. The cell survival curve was fitted using a single-hit multitarget model using GraphPad Prism 8 software (GraphPad Software, LLC, San Diego, USA). Sensitization enhancement ratio (SER) was expressed as an enhancement ratio determined at a survival fraction (SF) of 37%.

### 4.4. RNA Extraction and RT-qPCR Assay

The expression levels of common differentially expressed genes were measured by RT-qPCR in vitro. Total cellular RNA was extracted using TRIzol reagent (Invitrogen, San Diego, CA, USA). Total RNA (1 μg) was reverse transcribed into cDNA using a FastKing RT Kit (with gDNase) (Tiangen Biotechnology, Beijing, China). RT-qPCR was performed using SuperReal PreMix Plus (SYBR Green) (Tiangen Biotechnology, Beijing, China) in 20 μL reaction reagents using the MX3000P platform according to the manufacturer’s protocol. The primers used in the RT-qPCR assays are listed in Appendix A.

### 4.5. Western Blot Assay

Briefly, total cellular proteins were extracted with RIPA buffer (Beyotime Biotechnology, Shanghai, China) with protease inhibitor to obtain protein according to the manufacturer’s instruction. Then an equal amount of protein was separated by sodium dodecyl sulfate polyacrylamide gel electrophoresis (SDS-PAGE) on a 10% gel and transferred to a polyvinylidene difluoride (PVDF) membrane (Millipore, Bedford, MA, USA). The membrane was blocked with 5% nonfat milk in Tris-buffered saline/Tween 0.05% (TBST) for 2 h and incubated overnight at 4 °C with primary antibodies of anti-H3K27ac antibody (1:1000, ab177178, Abcam Inc., Shanghai, China), anti-β-actin antibody (1:1000, ab8226, Abcam Inc., Shanghai, China), anti-MMP14 antibody (1:1000, ab51074, Abcam Inc., Shanghai, China), and anti-H3 antibody (1:1000, #4499, Cell Signaling Technology Inc., Shanghai, China). After being washed with TBST, the membrane was incubated with secondary antibodies (1:5000, Signalway Antibody, College Park, MD, USA). Proteins in the membrane were detected using an ECL kit (Millipore, St. Louis, MO, United States), and band images were analyzed using the Bio-Rad ChemiDoc XRS system.

### 4.6. Cell Proliferation Assay

Cell proliferation was measured using the CCK-8 assay (Dojindo Laboratories, Kumamoto, Japan) according to the manufacturer’s instructions. Briefly, cells were seeded in 96-well plates at the appropriate concentration and cultured at 37 °C in an incubator for 4 h. When cells had adhered, 10 μL of CCK-8 working buffer was added to the 96-well plates and incubated at 37 °C for 2 h. The optical density at 450 nm was measured using a microplate reader (Tecan Infinite M200 Pro, Männedorf, Switzerland) to evaluate the cell proliferation rate. The CCK-8 assay for cell proliferation was repeated three times with three replicates for each cell sample.

### 4.7. RNA-Seq Data Download, Processing, and Analysis

Sequence data of GSE153982 were obtained from the Sequence Read Archive (SRA) and extracted using fastq-dump software (v2.10.9, https://github.com/rvalieris/parallel-fastq-dump, accessed 14 January 2021). Reads containing adapters, poly-N, and low-quality reads were removed with Trim Galore software (v0.6.6, https://github.com/FelixKrueger/TrimGalore, accessed 15 January 2021). Clean reads were mapped to the human reference genome (hg38) using TopHat software (v2.1.1, https://ccb.jhu.edu/software/tophat/index.shtml, accessed 19 January 2021) and Cufflinks software (v2.2.1, https://github.com/cole-trapnell-lab/cufflinks, accessed 21 January 2021). featureCount software (v2.0.1, http://subread.sourceforge.net, accessed 5 March 2021) was used to obtain read counts/fragments per kilobase of transcript per million fragments mapped (FPKM)/transcripts per million (TPM) for each noticed gene. The Bioconductor package limma (v3.36.0, https://bioconductor.org/packages/limma, accessed 6 March 2021) [44] was used to analyze the quantitative differentiation between the two identified groups. The Bioconductor package clusterProfiler (v3.18.1, https://bioconductor.org/packages/clusterProfiler/, accessed 27 May 2021) [45] was used to process the GSEA analysis, KEGG pathway enrichment, and GO terms enrichment.

### 4.8. ChIP-Seq Data Download, Processing, and Analysis

Sequence data of GSE131956 were obtained from the SRA. Unmapped reads were trimmed using Trim Galore v0.6.6. Alignment files were sorted and indexed using Bowtie2 software (v2.2.5, http://bowtie-bio.sourceforge.net/bowtie2/index.shtml, accessed 3 February 2021) and Samtools software (v1.10, http://github.com/samtools/samtools, accessed 3 February 2021). We defined the peaks using MACS2 software (v2.1.4, https://pypi.python.org/pypi/MACS2, accessed 3 February 2021) [46]. Peak annotation, comparison, and visualization were performed using the Bioconductor package ChIPseeker (v1.26.2, https://bioconductor.org/packages/ChIPseeker/, accessed 3 March 2021) [26]. Functional enrichment analysis was performed using the Bioconductor package clusterProfiler (v3.18.1, https://bioconductor.org/packages/clusterProfiler/, accessed 27 May. 2021) [45].

### 4.9. Gene Expression Analysis of Clinical Database

The CGGA database (http://www.cgga.org.cn, accessed 30 March 2021) was used for this study. Data from mRNAseq_693 were used for survival-associated gene analyses. “Clinical Data” and “Expression Data” were downloaded. High-grade gliomas with WHO grade IV in the “mRNAseq_693” dataset were studied. Subjects with an overall survival < 30% (overall survival < 8.5 months) were classified as having short-term survival, whereas those with an overall survival > 70% (overall survival > 20 months) were classified as having long-term survival. Differential expression analysis of the two groups was performed using the Bioconductor package limma.

### 4.10. RNA Interference and Drug Treatment

U251, U251R, and T98G cells were transfected with MMP14 siRNA (siMMP14) using riboFECT CP Transfection Agent (RiboBio, Guangzhou, China) according to the manufacturer’s protocol. The sequences of siRNAs are listed as follows: siMMP14 (5′ GGT CTC AAA TGG CAA CAT A 3′). The negative control siRNA had a random sequence. The transduction efficiency was consistently between 90% and 95%. U251, U251R, and T98G cells were treated with 5 mM SAHA for 24 h.

### 4.11. Statistical Analysis

All experiments were repeated at least three times, and the data are presented as mean ± standard deviation (SD). The differences between indicated groups were evaluated by Student’s t-test or one-way analysis of variance (ANOVA) using GraphPad Prism 8 (GraphPad software, San Diego, CA, USA). RNA-seq analysis was performed using the Bioconductor package limma v3.36.0. Statistical significance was set at *p <* 0.05.

## Figures and Tables

**Figure 1 ijms-22-10403-f001:**
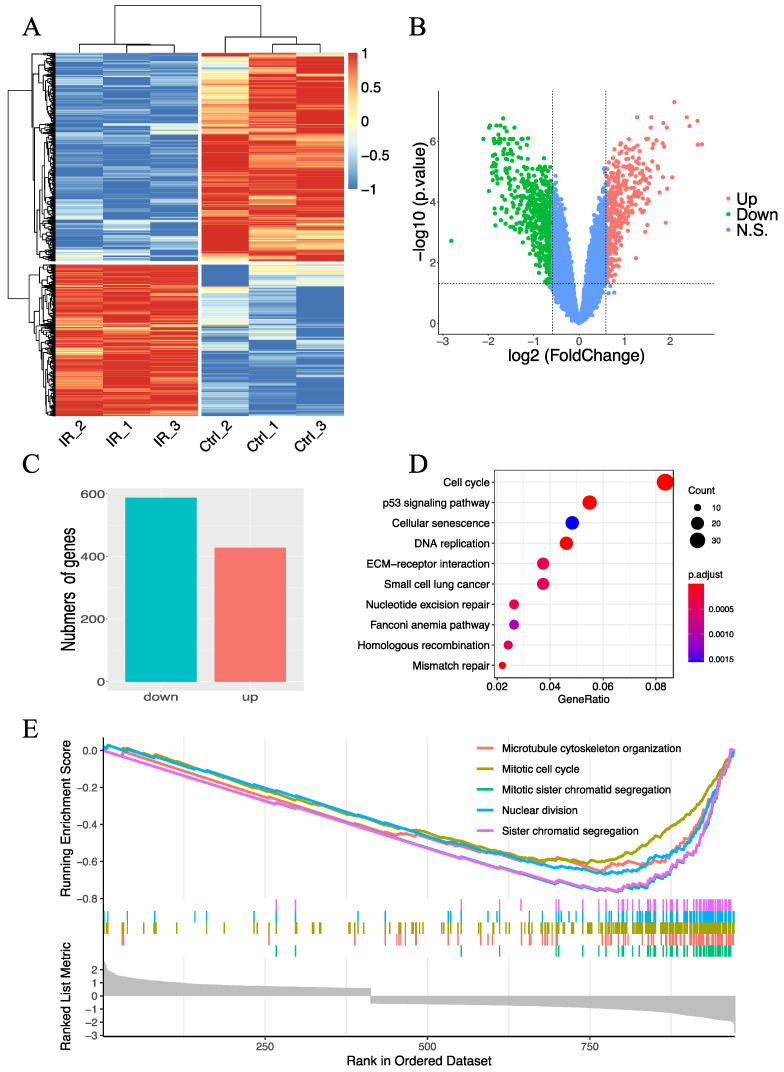
RNA-seq analysis of the paired irradiated and non-irradiated GBM cells in the GSE153982 dataset. (**A**) Heatmap of DEGs and normalized gene expression values were rescaled from –1 to 1. (**B**) Volcano plot of the genes. Red dots represent upregulated genes (Up), green dots represent downregulated genes (Down), and blue dots represent no significant change genes (N.S.). *p <* 0.05 is considered statistically significant with the threshold of fold change >1.5 or fold change <0.6667 (i.e., |log2 fold change} > 0.585). (**C**) Bar plot showing the numbers of downregulated genes (blue) and upregulated genes (red), respectively. (**D**) Top ten KEGG pathways enriched by the DEGs. (**E**) Top five gene sets enriched by DEGs based on GSEA. Abbreviations: DEG, differentially expressed gene; N.S., no significant change; KEGG, Kyoto Encyclopedia of Genes and Genomes; GSEA, gene set enrichment analysis.

**Figure 2 ijms-22-10403-f002:**
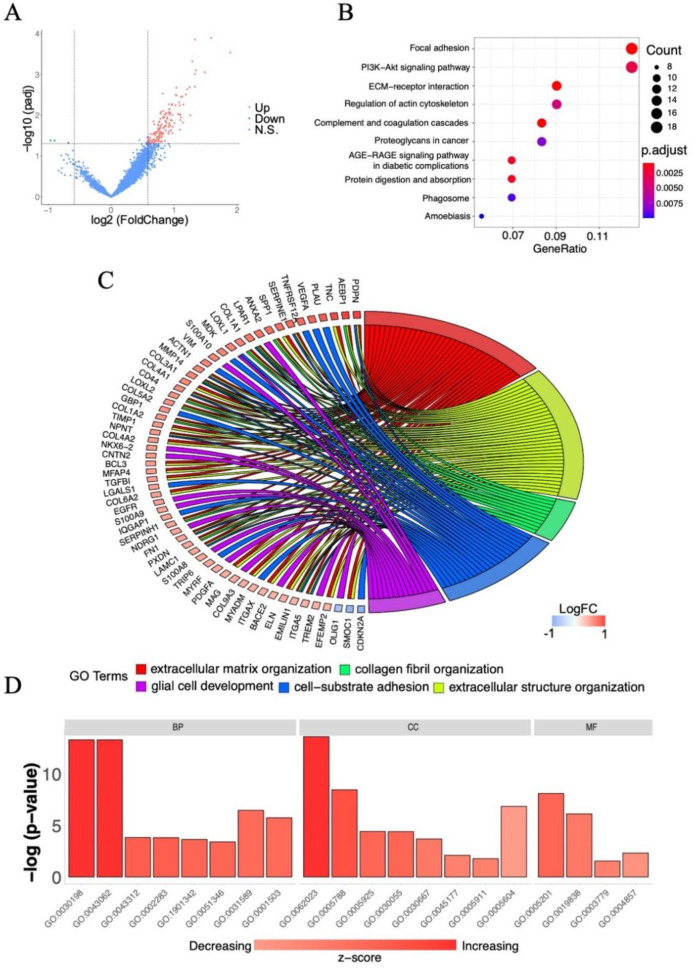
Analysis of the survival-associated genes from CGGA database. (**A**) Volcano plot of the distribution of gene expressions of high-survival group in comparison with low-survival group. Red dots represent upregulated genes (Up), green dots represent downregulated genes (Down), and blue dots represent no significant change genes (N.S.). *p <* 0.05 is considered statistically significant with the threshold of fold change >1.5 or fold change <0.6667 (i.e., |log2 fold change} > 0.585). (**B**) Dot plot of top ten KEGG pathways of survival-associated genes. (**C**) Chord plot of GO terms for survival-associated genes. logFC indicates log2 fold change. (**D**) Bar plot of GO terms for survival-associated genes. The z-score was automatically computed with the Bioconductor package GOplot. Abbreviations: CGGA, Chinese Glioma Genome Atlas; GO, Gene Ontology; FC, fold change; KEGG, Kyoto Encyclopedia of Genes and Genomes.

**Figure 3 ijms-22-10403-f003:**
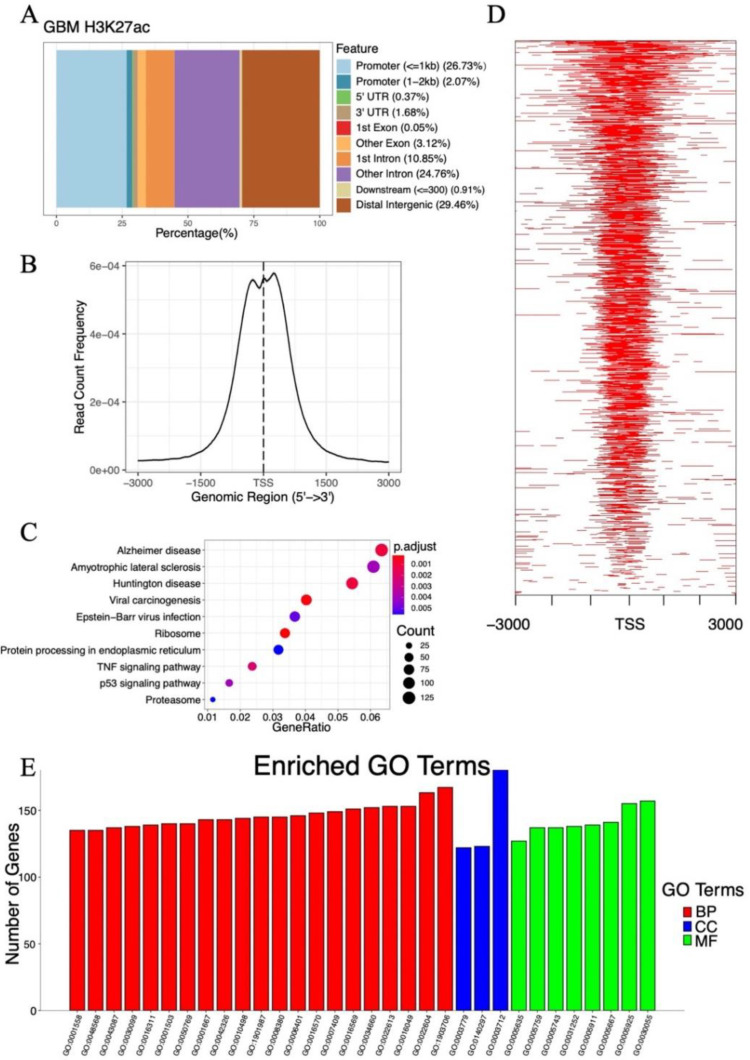
GSE131956 dataset analysis for the histone acetylation-controlled genes. (**A**) Distribution of peak locations including promoter, 5′ UTR, 3′ UTR, exon, intron, and intergenic region in the genome. (**B**) Correlation of global H3K27ac ChIP-seq peaks with TSSs. (**C**) Dot plot of top ten KEGG pathways of 5474 genes with H3K27ac peaks at their promoters. (**D**) Heatmap of the H3K27ac ChIP-seq peak at the TSS region of GBM cells. (**E**) GO analysis of 5474 genes with H3K27ac peaks at their promoters. Abbreviations: GBM, glioblastoma; UTR, untranslated region; H3K27ac, histone H3 lysine 27 acetylation; ChIP-seq, chromatin immunoprecipitation-sequencing; TSS, transcription start site; KEGG, Kyoto Encyclopedia of Genes and Genomes; GO, Gene Ontology; BP, biological process; CC, cellular component; MF, molecular function.

**Figure 4 ijms-22-10403-f004:**
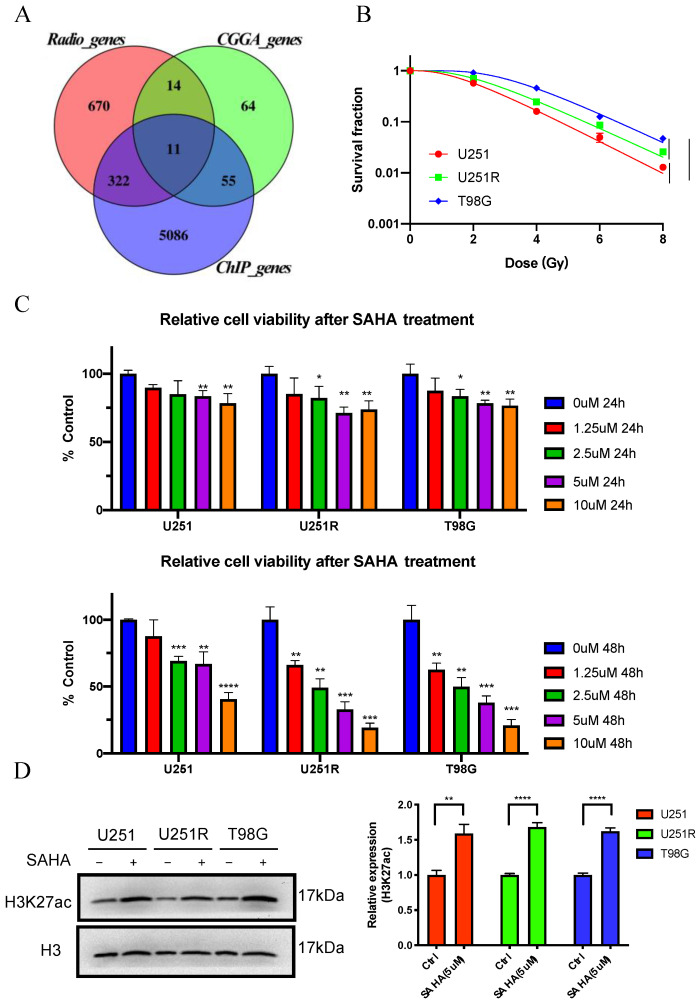
The effect of SAHA on H3K27ac levels in GBM cells. (**A**) Venn diagram illustrating the intersections of three candidate genes: “Radio_genes”, “CGGA_genes”, and “ChIP_genes”. (**B**) Colony formation assay of the survival curves of U251, U251R, and T98G cells irradiated with different doses of X-rays. (**C**) Cytotoxicity of SAHA treatment for 24 h and 48 h to three GBM cell lines. (**D**) Western blot assay of H3K27ac proteins in U251, U251R, and T98G cells after SAHA treatment. * *p <* 0.05, ** *p <* 0.01, *** *p <* 0.001, **** *p <* 0.0001. Abbreviations: CGGA, Chinese Glioma Genome Atlas; ChIP, chromatin immunoprecipitation; H3K27ac, histone H3 lysine 27 acetylation; TSS, transcription start site; SAHA, suberoylanilide hydroxamic acid; IR, irradiated; Ctrl, control.

**Figure 5 ijms-22-10403-f005:**
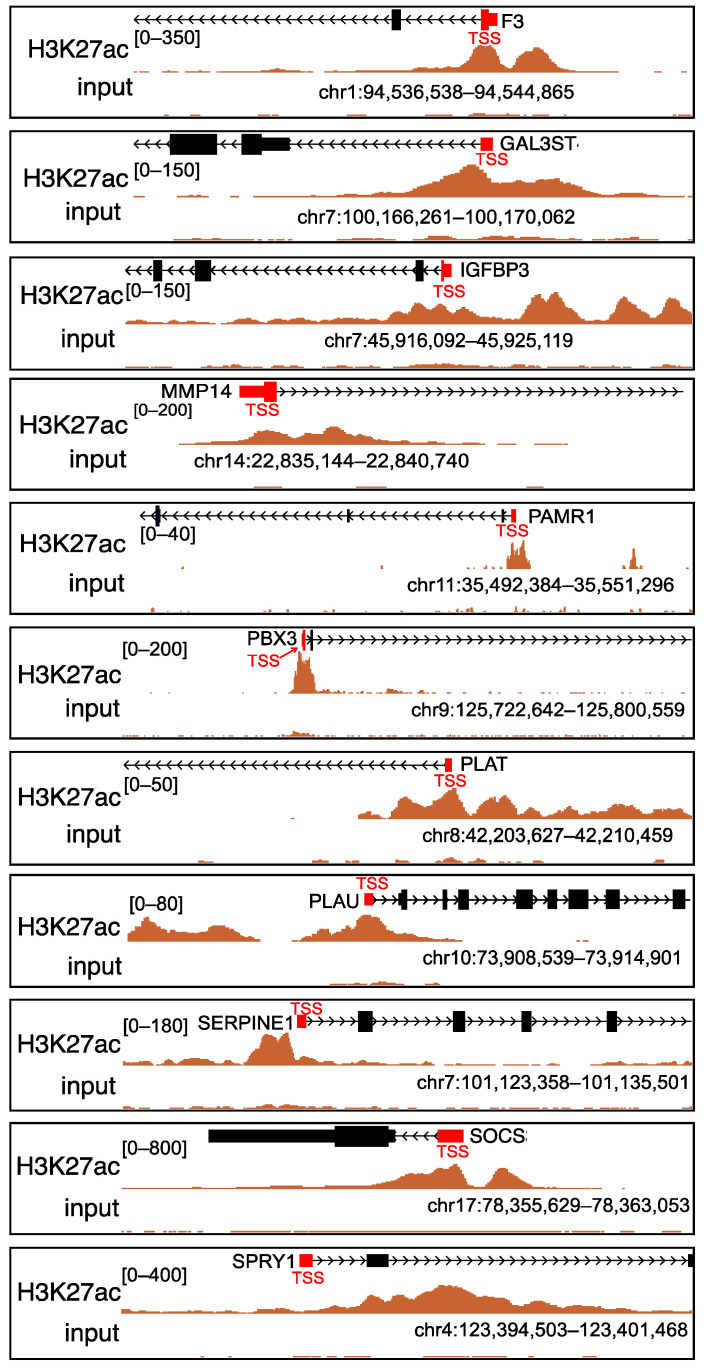
H3K27ac ChIP-seq peaks at TSSs in 11 candidate genes. Abbreviations: H3K27ac, histone H3 lysine 27 acetylation; ChIP-seq, chromatin immunoprecipitation-sequencing; TSS, transcription start site.

**Figure 6 ijms-22-10403-f006:**
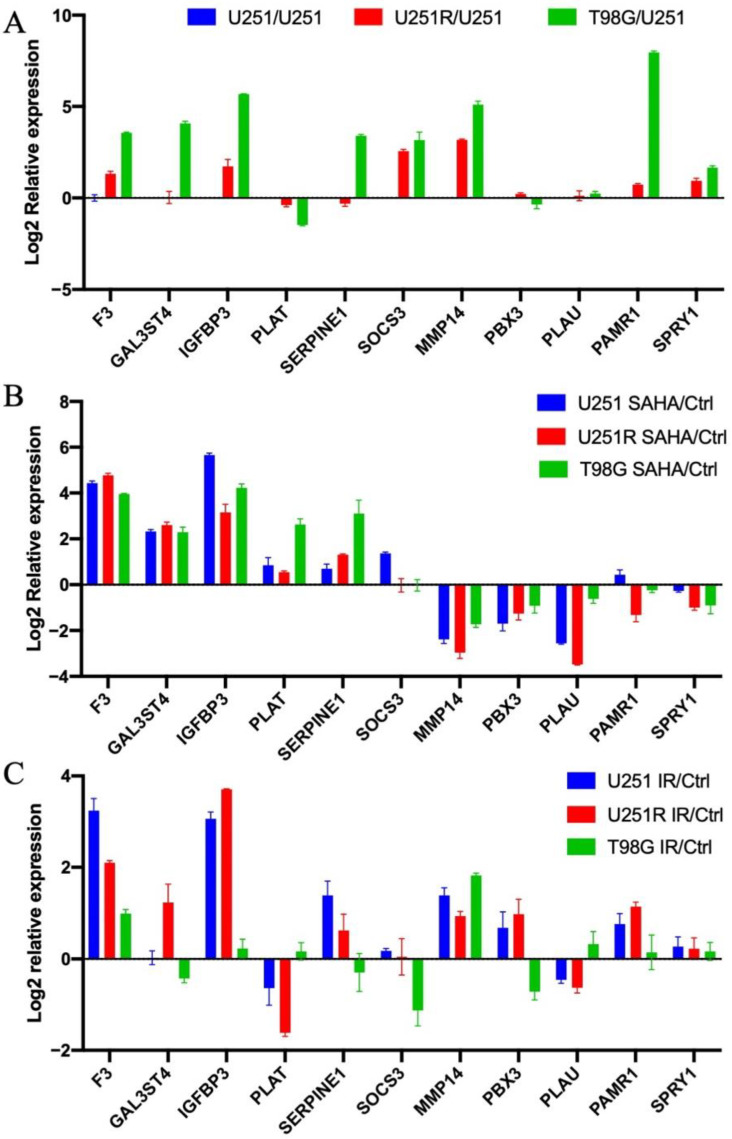
The RT-qPCR assay of eleven candidate genes. (**A**) RNA expression levels of 11 candidate genes in U251, U251R, and T98G cells. (**B**) RNA expression levels of 11 candidate genes in U251, U251R, and T98G cells after SAHA treatment. (**C**) The RNA expression levels of 11 candidate genes in U251, U251R, and T98G cells after irradiation for 48 h. The y-axis represents the log2 fold change of RNA expression level. Abbreviations: RT-qPCR, reverse transcription-quantitative polymerase chain reaction; SAHA, suberoylanilide hydroxamic acid; IR, irradiated; Ctrl, control.

**Figure 7 ijms-22-10403-f007:**
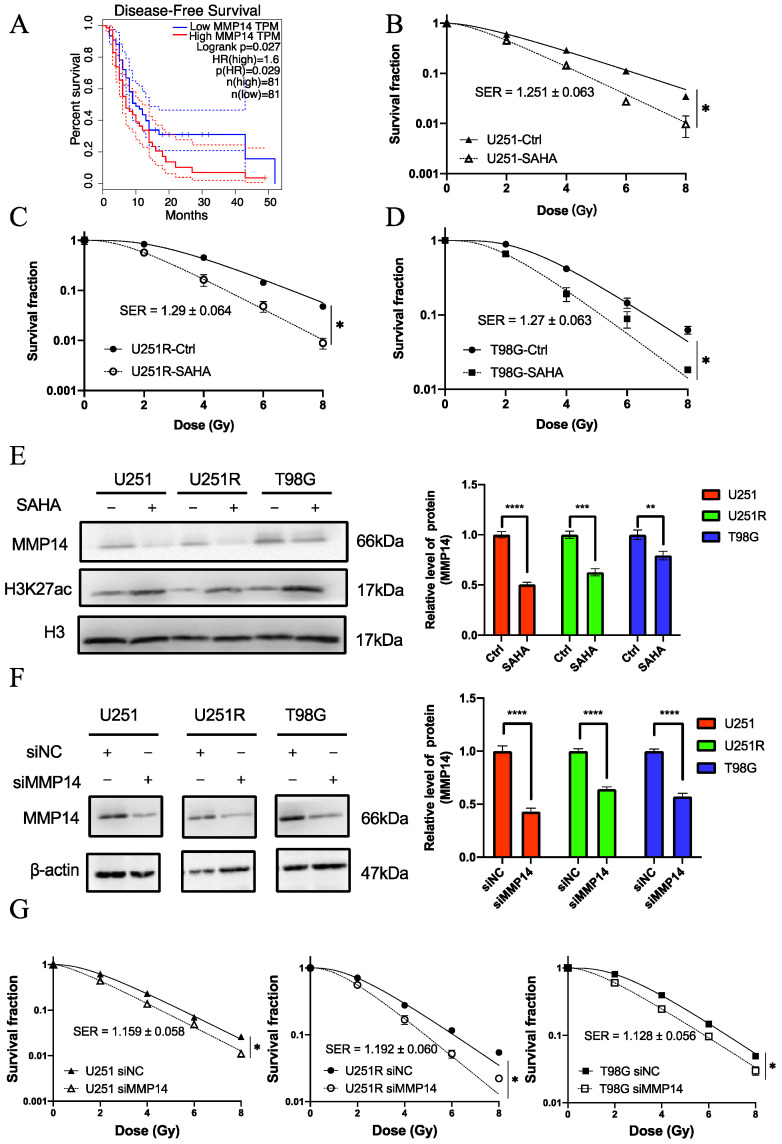
SAHA can increase the radiosensitivity of GBM by decreasing the expression of MMP14. (**A**) Kaplan–Meier analysis of the relationship between the expression of MMP14 in GBM and disease-free survival of the patients (data from GEPIA database). The solid line represents the survival curve of GBM patients with high (red) or low (blue) expression of MMP14. The dotted lines represent the 95% confidence intervals of two groups. “n” represents the number of patients in high or low expression group. (**B**–**D**) Colony formation assay of the survival curves of U251, U251R, and T98G cells irradiated with different doses of X-rays after SAHA treatment. (**E**) Western blot assay of MMP14 and H3K27ac proteins in U251, U251R, and T98G cells after SAHA treatment. (**F**) Western blot assay of MMP14 protein in U251, U251R, and T98G cells after siMMP14 transfection. (**G**) Colony formation assay of the survival curves of U251, U251R, and T98G cells irradiated with different doses of X-rays after siMMP14 transfection. * *p <* 0.05, ** *p <* 0.01, *** *p <* 0.001, **** *p <* 0.0001. Abbreviations: GBM, glioblastoma; GEPIA, Gene Expression Profiling Interactive Analysis; TPM, transcripts per million; HR, hazard ratio; SAHA, suberoylanilide hydroxamic acid; IR, irradiated; Ctrl, control; H3K27ac, histone H3 lysine 27 acetylation; SER, sensitization enhancement ratio; siMMP14, MMP14 siRNA; siNC, small interfering negative control.

**Table 1 ijms-22-10403-t001:** Eleven candidate genes of “CGGA_genes” and “Radio_genes”.

CGGA_genes			Radio_genes		
GeneID	Log_2_FC	*p*_Value	GeneID	Log_2_FC	*p*_Value
**PLAU**	1.18857781	0.00313738	PLAT	1.6308572	5.23E−09
**SERPINE1**	1.09935193	0.00977508	PLAU	1.26191627	0.00000183
**SOCS3**	1.03888471	0.00329858	SERPINE1	1.24394928	0.00000518
**F3**	0.92412421	0.00724404	MMP14	1.09278289	2.4E−09
**MMP14**	0.91836114	0.00541851	PAMR1	0.95922717	2.13E−07
**IGFBP3**	0.91328995	0.02870288	F3	0.80955228	0.00000134
**PAMR1**	0.77039761	0.01443068	GAL3ST4	0.76133027	0.000011
**PLAT**	0.76908966	0.02181263	IGFBP3	0.75038818	0.00000156
**SPRY1**	0.72918236	0.03779377	SPRY1	−0.6228777	0.00073747
**PBX3**	0.72507171	0.02756671	SOCS3	−0.7459958	0.0000168
**GAL3ST4**	0.61764614	0.04685508	PBX3	−1.0973208	0.00000114

Abbreviations: Log2FC, log2 fold change; PLAU, plasminogen activator, urokinase; SERPINE1, serpin family E member 1; SOCS3, suppressor of cytokine signaling 3; F3, coagulation factor III; MMP14, matrix metallopeptidase 14; IGFBP3, insulin-like growth factor binding protein 3; PAMR1, peptidase domain containing associated with muscle regeneration 1; PLAT, plasminogen activator, tissue type; SPRY1, sprouty RTK signaling antagonist 1; PBX3, PBX homeobox 3; GAL3ST4, galactose-3-O-sulfotransferase 4.

## Data Availability

The data presented in this study are available on request from the corresponding author upon reasonable request.

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
