# Peer review of "MMP14 Contributes to HDAC Inhibition-Induced Radiosensitization of Glioblastoma"

_ijms, 2021, doi:10.3390/ijms221910403_

Round 1

Reviewer 1 Report

The title “MMP14 contributes to HDAC inhibition-induced radiosensitization of glioblastoma” is appropriate and keywords represent the article adequately.

The paper presents the study of the mechanisms underlying the radiosensitizing effect of SAHA (an inhibitor of histone deacetylase) in GBM treatment. The authors analyzed some database and then the associated genes with survival. This work could be appreciate by scientific community but there are some integration to do.

Minor revision

  1. In the introduction better describe the function of MMP14 .
  2. In material and methods, the description of Western blot need to be improved.
  3. The authors must explain why selected the U251 cell line. Add some citations to improve the selected cell line.
  4. The authors should add some information and citations why the have selected the dose of 4 Gy
  5. Figure 2d is not clear  
  6. Figure 5: character formatting of caption is wrong.
  7. Figure 6: the captions of E and F panels are inverted and not clear .
  8. Line 299-305/306-312: the same paragraph is repeated twice.
  9. improve the quality and the statistical analysis in all the graph.
  10. Moreover, it would be useful to carry out a careful check of English grammar avoiding also spelling errors and reformulate some sentences in line 22-23, 32, 332…

Author Response

Please see the attachment for the response to reviewer.

-----

Responses to Reviewer comments

Reviewer 1

The title “MMP14 contributes to HDAC inhibition-induced radiosensitization of glioblastoma” is appropriate and keywords represent the article adequately.

The paper presents the study of the mechanisms underlying the radiosensitizing effect of SAHA (an inhibitor of histone deacetylase) in GBM treatment. The authors analyzed some database and then the associated genes with survival. This work could be appreciated by scientific community but there is some integration to do.

Minor revision

  1. In the introduction better describe the function of MMP14.

Response: Thank you for your positive response to our work and the kind advice. We have made further description about the function of MMP14 in the introduction (line 31-39 of page 2).

  1. In material and methods, the description of Western blot needs to be improved.

Response: Thank you for your critical comments on this point. We have improved the description of Western blot and the information of antibodies is added (line 36-43 of page 14).

  1. The authors must explain why selected the U251 cell line. Add some citations to improve the selected cell line.

Response: Thank you for your comments. We have further explained the reason why we choose U251 and T98G cell lines (line 7-11 of page 8 and line 1-2 of page 9). U251 and T98G cell lines with various radiosensitivity are widely used in the study of human GBM. The U251R cell line was previously established using fractionated irradiation of X-rays in our laboratory. T98G cells had the highest radioresistance among these cell lines, followed by U251R and U251 cells. We suggested that the high radioresistant cells might represent a poor clinical prognosis to some extent.

  1. The authors should add some information and citations why they have selected the dose of 4 Gy.

Response: Thank you for your positive and constructive comments on our submission. In the GEO dataset GSE153982 we applied, the RNA samples were extracted 48 hours after 4Gy irradiation. To match this dose, we treated GBM cell lines with same dose of 4 Gy in this study. We have added this explanation in the manuscript (line 34 of page 9 and line 1-2 of page 10).

  1. Figure 2d is not clear  

Response: We have improved the quality of Figure 2D.

  1. Figure 5: character formatting of caption is wrong.

Response: We found that due to the Office version, the picture was not shown properly. We have improved the caption of Figure 5. (line 15 of page 10) (Figure 6 in the revised manuscript).

  1. Figure 6: the captions of E and F panels are inverted and not clear.

Response: We have improved the explanation of these panels. (Figure 7 in the revised manuscript).

  1. Line 299-305/306-312: the same paragraph is repeated twice.

Response: We have deleted the same paragraph.

  1. improve the quality and the statistical analysis in all the graph.

Response: We have improved the quality and statistical analysis in all the graph and make it more readable.

  1. Moreover, it would be useful to carry out a careful check of English grammar avoiding also spelling errors and reformulate some sentences in line 22-23, 32, 332…

Response: Thank you for your comments. We have checked the English grammar throughout the manuscript, for example, have replaced “and but" to "and" (line 27 of page 1), “median survival duration” to “median survival time” (line 37 of page 1). To prevent ambiguity, we delete the sentences “U251R, a radioresistant human GBM cell line, was previously constructed and maintained in our laboratory.”

Reviewer 2 Report

In this study the authors propose to investigate how SAHA (an inhibitor of histone deacetylase) can improve the prognosis of glioblastoma (GBM) patients in combination with chemo/radiotherapy. To identify novel targets of suberoylanilide hydroxamic acid (SAHA), they first referred to many databases in order to examine which genes modulate their expression after irradiation. Second, they analyzed genes associated with survival and whose expression may be deregulated after SAHA treatment. Then, they evaluated the RNA expression level of eleven candidate genes of low-survival groups using three GBM human cell lines (U251, U251R, T98G) after the treatment with SAHA or irradiation. In conclusion, they focused their attention on the evaluation of MMP14 expression after SAHA treatment and irradiation.

The topic is interesting and of relevant scientific interest, because of targeting specific biomarkers to radiosensitize glioblastoma has been acquiring a key role to suggest new therapies; moreover, the evaluation with a genetic-molecular approach may offer a great contribution to understand the pathophysiological mechanism.

However, it is mandatory to make some modifications and integrations to improve the quality of the manuscript.

Here some suggestions

  1. In the introduction, the function of histone deacetylase and its inhibitor is not fully described. The authors should also mention the role and the function of MMP14.
  2. In Materials and methods some information should be more detailed:
  • Western blot description is approximative, indeed Ab species, isotype, clonality and host of the antibodies used are missed.
  • The use of U251R a radioresistant human GBM cell line is an important point. The authors should add the explanation why this cell line is considered radioresistant even if it is briefly explained the procedure to obtain it.
  1. Results require several corrections and the figures require major improvements:
  • A weak point is the evaluation of biological effects induced by MMP14 and SAHA in GBM. The only functional experiment was the clonogenic assay but MMP14 not only promotes tumor cell proliferation but also plays a role in GBM invasion, being engaged in the cleavage and proteolysis of several proteins that have adhesion functions (also in epithelial-mesenchymal transition EMT). Therefore, the authors should also test migration and/or invasion effects on GBM cells after irradiation/MMP14inhibtion/SAHA treatment.
  • The authors should explain the choice of 4 Gy dose radiation in GEO dataset GSE152982 analysis and timing (48 hrs after irradiation).
  • Figure 2D: graph’s description is unreadable and wrong characters
  • Please improve the quality with appropriate characters and dimensions in all figures.
  • Figure 3A: some regions of the gene indicated both in the legend and caption (such as 5′UTR region) are missing in the figure.
  • Statistical analysis are missed in Figure4C and D.
  • In addition to statistical analysis of survival curves, the authors should also provide radiobiological parameters such as alfa, beta, alfa/beta ratio and dose modifying factor to evaluate synergistic or additive effect of siMMP14 and SAHA combined with irradiation. These would also allow to compare the survival effects between SAHA and siMMP14 combined with irradiation.
  • The authors decided to investigate the role of MMP14 after the validation of 11 genes expression analysis in basal, irradiated and SAHA conditions. Why the RNA expression level of these 11 candidate genes were not evaluated after SAHA treatment combined with irradiation?
  • Maybe it would be more organized to merge tables 1 and 2 into one table?
  1. Discussion needs to be expanded. Particularly, the authors should elucidate their results comparing data with other in literatures about MMP14 inhibition or SAHA combined with radiotherapy. Moreover, the authors should discuss the significance of optimizing the synergistic molecular targeted drugs and radiation-based therapies for GBM. For this purpose, the authors may check and cite this paper: Torrisi F. et al., The Role of Hypoxia and SRC Tyrosine Kinase in Glioblastoma Invasiveness and Radioresistance. Cancers (Basel). 2020 Oct 4;12(10):2860. doi: 10.3390/cancers12102860. PMID: 33020459; PMCID: PMC7599682.
  2. A general check of the text by a native English speaker would be useful, also improving the construction of some sentences and avoiding spelling mistakes. Some examples:
  • Lines 22-23. “MMP14 mRNA was considerably highly expressed in the radioresistant cell lines and but was reduced by SAHA treatment”.
  • Line 32: “the median survival duration”. You’d better write median survival time or expectancy. Line 51: “In some individuals”: it would be better to use “patients”.
  • Lines 332-333: U251R, a radioresistant human GBM cell line, was previously constructed and maintained in our laboratory. What exactly does it mean? “constructed”? (See also this point in the revisions of materials and methods).
  • Check and write the full name of each abbreviation in the abstract.
  • Figure 5: character formatting of caption is wrong.
  • Figure 6: maybe the explanations of of E and F panels are inverted.
  • Line 299-305/306-312: the same paragraph is repeated twice.

Author Response

Please see the attachment for the response to reviewer.

-----

Responses to Reviewer comments

Reviewer 2:

In this study the authors propose to investigate how SAHA (an inhibitor of histone deacetylase) can improve the prognosis of glioblastoma (GBM) patients in combination with chemo/radiotherapy. To identify novel targets of suberoylanilide hydroxamic acid (SAHA), they first referred to many databases in order to examine which genes modulate their expression after irradiation. Second, they analyzed genes associated with survival and whose expression may be deregulated after SAHA treatment. Then, they evaluated the RNA expression level of eleven candidate genes of low-survival groups using three GBM human cell lines (U251, U251R, T98G) after the treatment with SAHA or irradiation. In conclusion, they focused their attention on the evaluation of MMP14 expression after SAHA treatment and irradiation.

The topic is interesting and of relevant scientific interest, because of targeting specific biomarkers to radiosensitize glioblastoma has been acquiring a key role to suggest new therapies; moreover, the evaluation with a genetic-molecular approach may offer a great contribution to understand the pathophysiological mechanism.

However, it is mandatory to make some modifications and integrations to improve the quality of the manuscript.

Here some suggestions

  1. In the introduction, the function of histone deacetylase and its inhibitor is not fully described. The authors should also mention the role and the function of MMP14.

Response: Thanks for the reviewer’s positive responses and kind comments. We have made further description about the function of MMP14 (line 31-39 of page2), histone deacetylase and its inhibitor (line 3-7 of page 2) in the introduction.

  1. In Materials and methods some information should be more detailed:
  • Western blot description is approximative, indeed Ab species, isotype, clonality and host of the antibodies used are missed.

Response: We have improved the description of Western blot and the information of antibodies is added. (line 36-43 of page 14).

  • The use of U251R a radioresistant human GBM cell line is an important point. The authors should add the explanation why this cell line is considered radioresistant even if it is briefly explained the procedure to obtain it.

Response: Thank you for your constructive comments. We have further explained the reason why U251R is considered radioresistant (line 10-11 of page 8 and line 1-2 of page 9). The U251R cell line was previously established from U251 using fractionated irradiation of X-rays in our laboratory, which shows increased survival after irradiation in compared with U251.

  1. Results require several corrections and the figures require major improvements:
  • A weak point is the evaluation of biological effects induced by MMP14 and SAHA in GBM. The only functional experiment was the clonogenic assay but MMP14 not only promotes tumor cell proliferation but also plays a role in GBM invasion, being engaged in the cleavage and proteolysis of several proteins that have adhesion functions (also in epithelial-mesenchymal transition EMT). Therefore, the authors should also test migration and/or invasion effects on GBM cells after irradiation/MMP14inhibtion/SAHA treatment.

Response: This is a great suggestion. Indeed, much deep investigations are still required to know the functions of MMP14 in GBM cells. The role of MMP14 in migration and invasion of GBM cells has been confirmed previously (we have added some references in the Introduction, (line 31-39 of page 2). Therefore, this study just focused on the expression of MMP14 in SAHA treated GBM cells by bioinformatics analysis. However, we are willing to further investigate the mechanism of MMP14 in radioresistance, including EMT, in the future.

  • The authors should explain the choice of 4 Gy dose radiation in GEO dataset GSE152982 analysis and timing (48 hrs after irradiation).

Response: In the GEO dataset GSE153982 we applied, the RNA samples were extracted 48 hours after 4Gy irradiation. To match this dose and analysis time, we treated GBM cell lines with same dose of 4 Gy in this study. We have added this explanation in the manuscript (line 34 of page 9 and line 1-2 of page 10).

  • Figure 2D: graph’s description is unreadable and wrong characters

Response: We have improved the quality of Figure 2D.

  • Please improve the quality with appropriate characters and dimensions in all figures.

Response: We have improved the quality with appropriate characters and dimensions in the figures and make it readable.

  • Figure 3A: some regions of the gene indicated both in the legend and caption (such as 5′UTR region) are missing in the figure.

Response: It is too small to be shown in the figure, so we have shown the percentage in the legend.

  • Statistical analysis are missed in Figure4C and D.

Response: We have added the statistical analysis in the figures. In addition, we have re-plotted original Figure 4D to Fig. 4D and E in the revised manuscript to show the statistical difference between different treatments.   

  • In addition to statistical analysis of survival curves, the authors should also provide radiobiological parameters such as alfa, beta, alfa/beta ratio and dose modifying factor to evaluate synergistic or additive effect of siMMP14 and SAHA combined with irradiation. These would also allow to compare the survival effects between SAHA and siMMP14 combined with irradiation.

Response: We have used the SER (Sensitization enhancement ratio) to compare the survival effects between SAHA and siMMP14 combined with irradiation.

  • The authors decided to investigate the role of MMP14 after the validation of 11 genes expression analysis in basal, irradiated and SAHA conditions. Why the RNA expression level of these 11 candidate genes were not evaluated after SAHA treatment combined with irradiation?

Response: We are grateful for this constructive comment. This part of work is mainly to verify the results of bioinformatics analysis. The validation of the RNA expression level of these 11 candidate genes after SAHA treatment combined with irradiation may not support the accuracy of our bioinformatics analysis. The GEO dataset GSE153982 was examined to assess radiation-induced changes in gene expression in GBM. The CGGA database was analyzed to screen genes associated with survival. The GEO dataset GSE131956 was analyzed to screen genes whose expression may be altered after SAHA treatment. To further verify the accuracy of bioinformatic analysis, we investigated the validation of 11 genes expression in basal, irradiated and SAHA treatment. In this study, we did not focus on the changes of RNA expression level after SAHA combined with radiation, but are centered on the screen of the gene which cause radioresistance. Besides, due to lack of relevant database, the database about the RNA expression level after SAHA treatment combined with irradiation was not analyzed in our study. For the above reasons, we did not perform these experiments.

  • Maybe it would be more organized to merge tables 1 and 2 into one table?

Response: Thank you for the comment. We have organized table 1 and table 2 into one table.

  1. Discussion needs to be expanded. Particularly, the authors should elucidate their results comparing data with other in literatures about MMP14 inhibition or SAHA combined with radiotherapy. Moreover, the authors should discuss the significance of optimizing the synergistic molecular targeted drugs and radiation-based therapies for GBM. For this purpose, the authors may check and cite this paper: Torrisi F. et al., The Role of Hypoxia and SRC Tyrosine Kinase in Glioblastoma Invasiveness and Radioresistance. Cancers (Basel). 2020 Oct 4;12(10):2860. doi: 10.3390/cancers12102860. PMID: 33020459; PMCID: PMC7599682.

Response: We have further elucidated our results comparing data with other in literatures about MMP14 inhibition (line 26-29 page 13) and SAHA (line 32-36 page 13) combined with radiotherapy. We have cited this paper (ref. 35) and discussed the significance of optimizing the synergistic molecular targeted drugs and radiation-based therapies for GBM. (line 26-28 page 11)

  1. A general check of the text by a native English speaker would be useful, also improving the construction of some sentences and avoiding spelling mistakes. Some examples:
  • Lines 22-23. “MMP14 mRNA was considerably highly expressed in the radioresistant cell lines and butwas reduced by SAHA treatment”.

Response: We have changed “and but" to "and". (line 27 of page 1).

  • Line 32: “the median survival duration”. You’d better write median survival time or expectancy.

Response: We have changed “median survival duration” to “median survival time” (line 37 of page 1).

  • Line 51: “In some individuals”: it would be better to use “patients”.

Response: We have changed “individuals” to “patients” (line 15 of page 2 and line 8 of page 11)

  • Lines 332-333: U251R, a radioresistant human GBM cell line, was previously constructed and maintained in our laboratory. What exactly does it mean? “constructed”? (See also this point in the revisions of materials and methods).

Response: Thank you for your critical comments on this point. To prevent ambiguity, we delete these sentences.

  • Check and write the full name of each abbreviation in the abstract.

Response: We have added the full name of “HDAC” (line 16 of page 1), “SAHA” (line 17 of page 1), “GEO” (line 18-19 of page 1) and “CGGA” (line 22 of page 1) in the Abstract.

  • Figure 5: character formatting of caption is wrong.

Response: Thank you for your comments. We found that due to the Office version, the picture was not shown properly. We have improved the caption of Figure 5. (line 15 of page 10) (Figure 6 in the revised manuscript).

  • Figure 6: maybe the explanations of of E and F panels are inverted.

Response: Thank you for your critical comments on this point. We have improved the explanation of these panels. (Figure 7 in the revised manuscript).

  • Line 299-305/306-312: the same paragraph is repeated twice.

Response: We have deleted the repeated paragraph.

Reviewer 3 Report

General evaluation:

In their study, Zhou and co-workers combined three bioinformatic analyses to identify 11 genes, associated with glioblastoma (GBM) patients’ survival, activated by irradiation and regulated by the HDAC inhibitor SAHA. In more detail, GEO datasets were analyzed for radiation-induced gene expression in GBM and changes induced by SAHA treatment and the CGGA database was screened for survival-associated genes in GBM patients. The intersection of these analyses yielded 11 genes, which regulation upon SAHA treatment or irradiation was validated by quantitative PCR. Of these genes, Matrix metalloproteinase 14 (MMP14) expression was induced upon irradiation but decreased after SAHA treatment. The impact of MMP14 expression on clonogenic survival was further investigated by siRNA knockdown, showing a radiosensitizing effect in three GBM cell lines. The authors concluded from their findings, that SAHA-mediated MMP14 downregulation contributes to radiosensitization of GBM.

Overall, the manuscript is well written, deals with an important and interesting topic and results are clearly arranged and mostly comprehensibly presented. Findings, derived from bioinformatics are confirmed and validated in three cell lines, which enhances the significance of the study.

There are, however, points of criticism which reduces the enthusiasm for the study and which need to be addressed as listed below.

Points of criticism:

  1. Results section, p. 4, lines 101-103: What does the sentence: „Subjects with 101 an overall survival <30% were classified as having low survival, whereas those with an 102 overall survival >30% were classified as having high survival“ mean? This needs to be explained in more detail: how were the patients exactly stratified?
  2. Legend to Figure 1: Please introduce the term „GSEA“ at first usage in the Figure legends.
  3. Results section, p. 4, line 108: Please introduce the term „ECM“ at first appearance in the text.
  4. Figure 2D: Please improve the y-axis and legend labeling, it is not readable.
  5. Figure 4B: Would it be possible to improve the readability of the labeling, because even at higher magnification it is hard to read?
  6. Results section, p. 7-8, lines 176-178: Where are the expression levels, described in the following sentence, displayed: „To further verify this conclusion, the expression of candidate genes was detected in 176 human GBM cell lines (U251 and T98G) and U251 homologous radioresistant cells“? It seems, that expression of candidate genes is shown in Figure 5, so it might be better to move the sentence to the next paragraph.
  7. Table 1: Please introduce the abbreviations used in the Table in the Table legend.
  8. Figure 5: Please improve the labeling of the x-axis, e.g., MMP14 reads „MMP1 4“, SERPINE1 reads „SER PI NE1“ and so on.
  9. Figure 6B, C, D, G: Are there any significant changes between control and SAHA-treated cells?
  10. Figures 4C, D, E; 5; 6B, C, D, E, F, G: Please indicate the number of independent experiments performed to obtain the data displayed.
  11. Discussion section, p. 12, line 273: In the sentence „Containing RNA-seq data of the irradiated GBM cell line, GEO dataset GSE153982 …“, the authors refer to the GEO dataset investigated as containing ONE irradiated cell line. However, in Figure 1A, three corresponding pairs (IR_1 – Ctrl_1, IR_2 – Ctrl_2, IR_3 – Ctrl_3) are displayed. Please describe the data sets in more detail. Which cell line(s) have been investigated?
  12. Discussion section, p. 13, 319-323: The following sentences are not intellegible: „According to Kawamata et al., SAHA can inhibit gene expression by disrupting the PI3K/Akt pathway [30]. Additionally, the Akt signaling pathway has been shown to repress MMP14 expression in various cancer cell lines [31]. Therefore, we speculate that SAHA may repress the expression of MMP14 by blocking the PI3K/Akt pathway.“ How can blocking of the PI3K/Akt pathway repress the expression of MMP14 if Akt signaling represses MMP14 expression? Please clarify.
  13. Material and Methods section: Please provide the manufacturer of SAHA and the solvent used to reconstitute the substance. Were the controls treatedalso with equal amounts of solvent? Please specify.

Author Response

Please see the attachment for the response to reviewer.

-----

Responses to Reviewer comments

Reviewer 3:

General evaluation:

In their study, Zhou and co-workers combined three bioinformatic analyses to identify 11 genes, associated with glioblastoma (GBM) patients’ survival, activated by irradiation and regulated by the HDAC inhibitor SAHA. In more detail, GEO datasets were analyzed for radiation-induced gene expression in GBM and changes induced by SAHA treatment and the CGGA database was screened for survival-associated genes in GBM patients. The intersection of these analyses yielded 11 genes, which regulation upon SAHA treatment or irradiation was validated by quantitative PCR. Of these genes, Matrix metalloproteinase 14 (MMP14) expression was induced upon irradiation but decreased after SAHA treatment. The impact of MMP14 expression on clonogenic survival was further investigated by siRNA knockdown, showing a radiosensitizing effect in three GBM cell lines. The authors concluded from their findings, that SAHA-mediated MMP14 downregulation contributes to radiosensitization of GBM.

Overall, the manuscript is well written, deals with an important and interesting topic and results are clearly arranged and mostly comprehensibly presented. Findings, derived from bioinformatics are confirmed and validated in three cell lines, which enhances the significance of the study.

There are, however, points of criticism which reduces the enthusiasm for the study and which need to be addressed as listed below.

Points of criticism:

  1. Results section, p. 4, lines 101-103: What does the sentence: „Subjects with 101 an overall survival <30% were classified as having low survival, whereas those with an 102 overall survival >30% were classified as having high survival“ mean? This needs to be explained in more detail: how were the patients exactly stratified?

Response: We have further explained how the patients were exactly stratified (line 13-15 of page 4). Subjects with an overall survival < 30% (overall survival < 8.5 months) were classified as having short-term survival, whereas those with an overall survival > 70% (overall survival > 20 months) were classified as having long-term survival.

  1. Legend to Figure 1: Please introduce the term „GSEA“ at first usage in the Figure legends.

Response: We have introduced the term “GSEA” in the Figure 1 legends. (line 8 of page 4)

  1. Results section, p. 4, line 108: Please introduce the term „ECM“ at first appearance in the text.

Response: We have written the full name of ECM (line 19 of page 4).

  1. Figure 2D: Please improve the y-axis and legend labeling, it is not readable.

Response: We have improved the quality of Figure 2D.

  1. Figure 4B: Would it be possible to improve the readability of the labeling, because even at higher magnification it is hard to read?

Response: We have improved the quality of Figure 4B and showed it as a new figure (Figure 5) in the revised manuscript.

  1. Results section, p. 7-8, lines 176-178: Where are the expression levels, described in the following sentence, displayed: „To further verify this conclusion, the expression of candidate genes was detected in 176 human GBM cell lines (U251 and T98G) and U251 homologous radioresistant cells“? It seems, that expression of candidate genes is shown in Figure 5, so it might be better to move the sentence to the next paragraph.

Response: We appreciate for the suggestion. We have improved the sentence. (line 7-10 of page 8)

  1. Table 1: Please introduce the abbreviations used in the Table in the Table legend.

Response: We have introduced the abbreviations used in the Table 1 legend (line 22-27 of page 9)

  1. Figure 5: Please improve the labeling of the x-axis, e.g., MMP14 reads „MMP1 4“, SERPINE1 reads „SER PI NE1“ and so on.

Response: We found that due to the Office version, the picture was not shown properly. We have improved the labeling of the x-axis of Figure 5 (Figure 6 in the revised manuscript).

  1. Figure 6B, C, D, G: Are there any significant changes between control and SAHA-treated cells?

Response: We have used the SER (Sensitization enhancement ratio) to compare the survival effects between SAHA and siMMP14 combined with irradiation and added the significant difference marker in the figure. (Figure 7 in the revised manuscript).

  1. Figures 4C, D, E; 5; 6B, C, D, E, F, G: Please indicate the number of independent experiments performed to obtain the data displayed.

Response: We have indicated the number of independent experiments performed to obtain the data (line 23-24 of page 14 for Colony formation assay and line 9-10 of page 15 for Cell proliferation assay)

  1. Discussion section, p. 12, line 273: In the sentence „Containing RNA-seq data of the irradiated GBM cell line, GEO dataset GSE153982 …“, the authors refer to the GEO dataset investigated as containing ONE irradiated cell line. However, in Figure 1A, three corresponding pairs (IR_1 – Ctrl_1, IR_2 – Ctrl_2, IR_3 – Ctrl_3) are displayed. Please describe the data sets in more detail. Which cell line(s) have been investigated?

Response: Thank you for your critical comments on this point. RNA samples with three replicates from GEO dataset GSE153982 were extracted 48 h after irradiation. We have described the data sets in more detail. (line 44 of page 2 and line 38 of page 11)

  1. Discussion section, p. 13, 319-323: The following sentences are not intellegible: „According to Kawamata et al., SAHA can inhibit gene expression by disrupting the PI3K/Akt pathway [30]. Additionally, the Akt signaling pathway has been shown to repress MMP14 expression in various cancer cell lines [31]. Therefore, we speculate that SAHA may repress the expression of MMP14 by blocking the PI3K/Akt pathway.“ How can blocking of the PI3K/Akt pathway repress the expression of MMP14 if Akt signaling represses MMP14 expression? Please clarify.

Response: We agree with this comment and have updated the sentences to “The PI3K/Akt signaling pathway has been shown to activate MMP14 expression and enhance cell migration and invasion” (line 40-42 of page 13).

  1. Material and Methods section: Please provide the manufacturer of SAHA and the solvent used to reconstitute the substance. Were the controls treated also with equal amounts of solvent? Please specify.

Response: Thank you for your helpful comments. We have specified the manufacturer of SAHA and the solvent. (line 11-16 of page 14) The controls were treated were with equal amounts of solvent as well.

Round 2

Reviewer 2 Report

In my opinion, the revised document is now appropriate and of a good standard. I have no other suggestions to give before its publication

Author Response

Thanks for the reviewer's support.